

# Validating Plutonium-239+240 as novel soil redistribution tracer - a
# comparison to measured sediment yield
Katrin Meusburger[1][*], Paolo Porto[2], Judith Kobler Waldis[3], Christine Alewell[3]
[1]Swiss Federal Institute for Forest, Snow and Landscape Research WSL, CH-8903, Birmensdorf,
Switzerland
[2]Dipartimento di Agraria, Università degli Studi Mediterranea di Reggio Calabria, Reggio Calabria,
Italy
[3]Environmental Geosciences, University of Basel, Switzerland
*corresponding author: katrin.meusburger@wsl.ch



## Abstract

Quantifying soil redistribution rates is a global challenge addressed with direct sediment measurements (e.g., traps), models and isotopic, geochemical and radiogenic tracers. The isotope of Plutonium, namely Pu-239+240, is a relatively new soil redistribution tracer in this challenge. Direct validation of Pu-239+240 as soil redistribution is, however, still missing. We used a unique sediment yield time series in Southern Italy, reaching back to the initial fallout of Pu-239+240 to verify Pu-239+240 as a soil redistribution tracer. Distributed soil samples (n=55) were collected in the catchment, and at potential undisturbed reference sites (n=22), Pu-239+240 was extracted, measured with ICP-MS and converted to soil redistribution rates. Finally, we used a Generalized Additive model (GAM) to regionalize soil redistribution estimates for the catchment. For the catchment sites, mean Pu-239+240 inventories were significantly reduced ($16.8 \pm 10.2$ Bq m$^{-2}$) compared to the reference inventory ($40.5 \pm 3.5$ Bq m$^{-2}$) indicating the dominance of erosion. Converting these inventory losses into soil erosion rates resulted in an average soil loss of $22.2 \pm$ SD 7.2 t ha$^{-1}$ yr$^{-1}$. The uncertainties of the approach stemmed mainly from the high measurement uncertainties of low-activity samples where samples have been bulked over depth. Therefore, we recommend taking incremental soil samples and extracting ~20g of soil. The geographic coordinates and the flow accumulation best described the spatial pattern of erosion rates in the GAM model. Using those predictors to upscale Pu-derived soil redistribution rates for the entire catchment resulted in an average on-site loss of 20.7 t ha$^{-1}$ yr$^{-1}$, which corresponds very well to the long-term average sediment yield of 18.7 t ha$^{-1}$ yr$^{-1}$ measured at the catchment outlet and to Cs-137 derived soil redistribution rates. Overall, this comparison of Pu-derived soil redistribution rates with measured sediment yield data validates Pu-239+240 as a suitable retrospective soil redistribution tracer.

## Graphical abstract



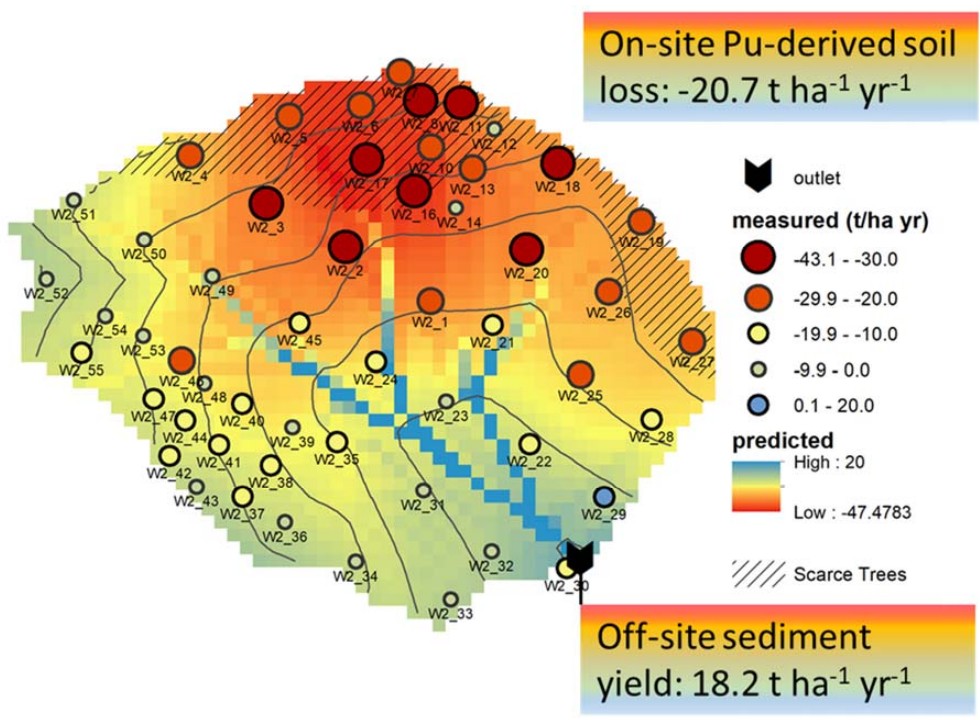

35

36



## 1   Introduction

Soil erosion endangers climate and food security and has considerable adverse off-site effects on freshwater systems (Reichstein et al., 2013; Amundson et al., 2015; Alewell et al., 2016; Panagos et al., 2016; Borrelli et al., 2017; Alewell et al., 2020). Plutonium isotopes, with their previous hazardous impacts on the environment and released as a product of thermonuclear weapons testing and from nuclear accidents (e.g. Chernobyl), may serve as a tool to quantify long-term soil loss (Alewell et al., 2017).

The approach to use Pu-239+240 as soil and sediment tracer is parallel to other fallout radionuclides (FRN) (Xu et al., 2015; Meusburger et al., 2018). Once deposited on the ground, FRNs strongly bind to soil particles and move across the landscape primarily through physical soil redistribution processes (IAEA, 2014). In this way, fallout radionuclides provide an effective and retrospective (since the time of the fallout) track of net soil and sediment redistribution (Zapata, 2003). However, Cs-137, the most commonly applied soil redistribution tracer, will reach its detection limit soon due to the successive decay (half-life = 30.17 years). Thus, alternative tracers like excess Pb-210 and Pu-239+240 have been explored (Wallbrink and Murray, 1996; Matisoff et al., 2002; Mabit et al., 2008; Kato et al., 2010; Porto et al., 2013; Teramage et al., 2015; Xu et al., 2015; Meusburger et al., 2018). While Pb-210 is associated with high uncertainties (Porto and Walling, 2012; Mabit et al., 2014; Meusburger et al., 2018), the characteristics of Pu-239+240 seem more promising for soil tracing (Alewell et al., 2017).

The advent of Pu-239+240 as a soil redistribution tracer was accelerated by the adoption of the less time-consuming (minutes instead of hours per sample) Inductively Coupled Plasma Mass spectrometry (ICP-MS). It was a door opener for using Pu-239+240 as a soil erosion tracer. The application of Pu-239+240 comes along with other advantages, such as i) reduced initial spatial variability at undisturbed, so-called reference sites (Alewell et al., 2014; Meusburger et al., 2016), ii) less preferential uptake by plants (Froehlich et al., 2016), iii) the possibility to assess the origin of the fallout by determining $^{240}$Pu to $^{239}$Pu atom ratios or Cs-137 to Pu-239+240 activity ratios (Ketterer et al., 2004; Xu et al., 2013; Meusburger et al., 2016; Meusburger et al., 2020), iv) considerably smaller soil sample volume needed for analysis, and v) no decline due to decay, which is of particular relevance for locations with low initial Cs-137 fallout such as the southern hemisphere (Tims et al., 2010). The potential of Pu-239+240 further convenes with the availability of the new conversion model "Modelling Deposition and Erosion rates with RadioNuclides (MODERN)", suitable for estimating soil redistribution rates by comparing reference with soil redistribution affected inventories with any FRN (Arata et al., 2016a; Arata et al., 2016b).

Several studies (Schimmack et al., 2001; Tims et al., 2010; Hoo et al., 2011; Lal et al., 2013; Michelotti et al., 2013; Xu et al., 2013; Xu et al., 2015; Meusburger et al., 2018) have highlighted $^{239+240}$Pu's suitability as a soil redistribution tracer. However, to date, direct validation efforts to compare on-site FRN-based soil erosion rates with off-site sediment yields have focused on other FRNs such as Cs-137



and excess $^{210}$Pb (Porto et al., 2001; Porto et al., 2003; Porto and Walling, 2012; Porto and Callegari,
2022). For $^{239+240}$Pu-derived soil redistribution rates, such a direct validation is not achieved yet, to the
best of our knowledge. This study aims to fill this gap by validating $^{239+240}$Pu-derived soil redistribution
rates with a long-term time series of measured catchment suspended sediment yields.

## 2   Materials and Methods

### 2.1   Study site and soil sampling

This study takes advantage of a unique long-term sediment yield monitoring catchment (W2, 1.38 ha)
located near Crotone in Calabria, Southern Italy (35 m a.s.l., 39°09′02″N, 17°08′10″E). The steep
catchment with a mean average slope of ca. 35% is located in the ephemeral headwaters of the larger
Crepacuore basin (Fig. 1). The geology of this area consists of Upper Pliocene and Quaternary materials
and produced soils with a clay loam texture with 14.6%, 49.2%, and 36.2% of sand, silt, and clay,
respectively. The catchment was never ploughed, but in 1968, *Eucalyptus occidentalis Engl.* was
planted and cut again in 1978 and 1990. The tree cover is partly patchy, with about 20% of the area on
south-facing slopes having discontinuous tree and grass cover. The climate is Mediterranean, with a
mean annual precipitation of ~670 mm, predominantly occurring from October to March.



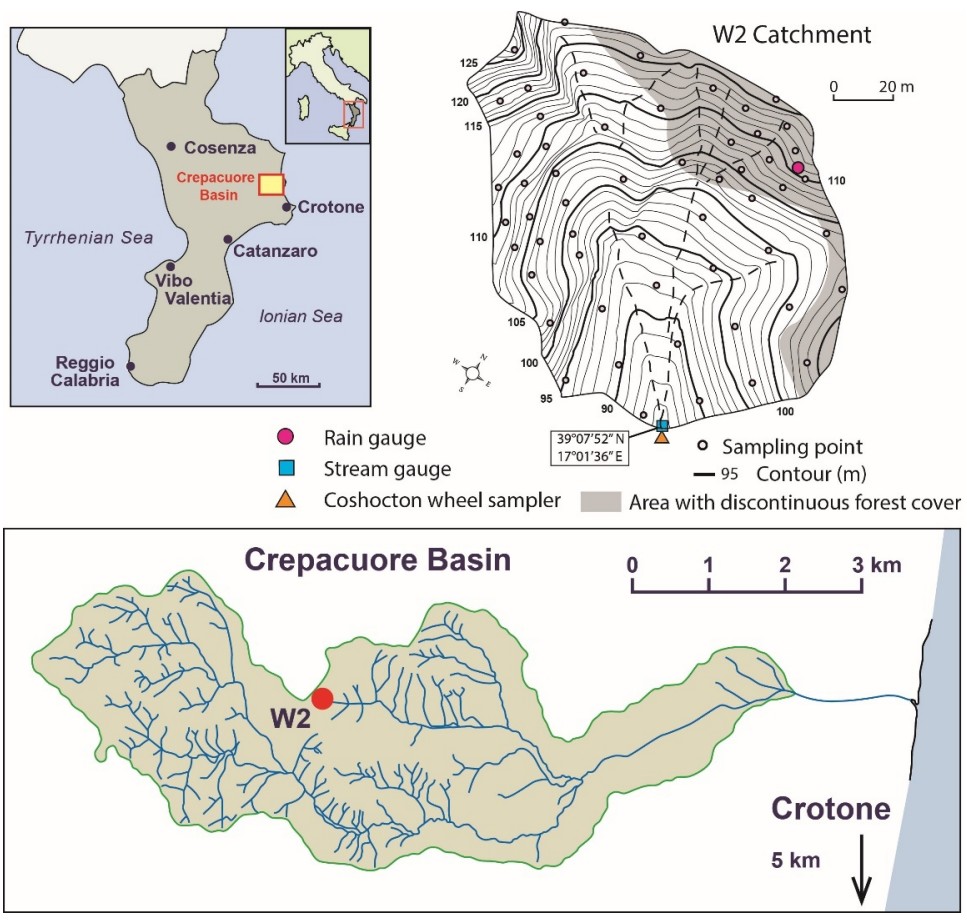

Fig. 1 Location of the studied headwater catchment W2 within the Crepacuore Basin (lower panel indicated by a red dot), Calabria, Italy.

In 2014, the collection of soil samples in the catchment was undertaken along an approximate 20 m × 20 m grid with additional cores collected from areas characterized by marked variability of vegetation cover or topography (Fig. 1). The samples were taken with a steel core tube (10 cm diameter) driven into the ground to a depth of 15 cm by a motorized percussion corer and subsequently extracted using a hand-operated winch. For each sampling point, two cores were taken, and they were bulked before analysis. This procedure provided a total of 55 composite bulk cores over the catchment area.

In March 2021, a new sampling campaign was undertaken to obtain information at the reference area to establish the baseline for $^{239+240}$Pu in the area. In this case, three depth profiles and nineteen additional bulk reference soil cores were collected in adjacent undisturbed rangeland with some scattered oaks (*Quercus pubescens*) at a similar altitude to the study catchment (see Porto and Callegari, 2022). The samples were collected using the same sampling device consisting of a motorized soil column cylinder



auger set in which a core tube (60 cm in length) with a larger internal diameter (11 cm) is accommodated. The three depth profiles were sectioned into increments of 2 cm and were analyzed separately for $^{137}$Cs and $^{239+240}$Pu content. Before radiometric analyses, all samples were dried and sieved to <2 mm. In a previous study, the soil samples collected within the catchment were analyzed for Cs-137 using high-resolution HPGe detectors available at the Agraria Department at the University Mediterranea of Reggio Calabria, Italy (Porto et al., 2014). Counting times for the samples collected during that campaign were ca. 80,000 s, providing a precision of ca. ±10% at the 95% confidence level. The reference samples of 2021 were also analyzed for Cs-137 with the same detector settings. All Cs-137 measurements were decay corrected to 01.01.2014 and used to calculate Pu-239+240 to Cs-137 activity ratios.

## 2.2 Extraction of Pu-239+240 and mass spectrometry for atom ratio and concentration measurements

All samples (5-10g) were oven-dried at 105°C for 48 h, mechanically disaggregated and dry-sieved to recover the <2 mm fraction. First, a representative sub-sample of this fraction was spiked with ~ 0.005 Bq of a $^{242}$Pu yield tracer (licensed solution from NIST). Next, Pu was leached with 16M nitric acid overnight at 80 °C and separated from the leach solution using a Pu-selective TEVA resin (Ketterer et al., 2011). The isotope dilution calculations determined the masses of $^{239}$Pu and $^{240}$Pu present in the sample and then converted them into the summed Pu-239+240 activity. The analysis was done with a Thermo X7 quadrupole ICP-MS system at Universidad de Cádiz. Please refer to Meusburger et al. (2020) for details on the instrument method.

## 2.3 Conversion of Pu-239+240 activities to soil redistribution rates

The total inventory (Bq m$^{-2}$) of each bulk soil core was calculated as the product of the measured Pu-239+240 activity (Bq kg$^{-1}$) and the dry mass of the <2 mm fraction of the bulk core (kg), divided by the surface area associated with the soil core (m$^2$). The inventories were converted into soil redistribution rates using the Profile Distribution model PDM (Walling et al., 2002; Walling et al., 2014) and the model Modelling Deposition and Erosion rates with RadioNuclides (MODERN (Arata et al., 2016a; Arata et al., 2016b)). The profile distribution model is commonly employed to interpret the shape of an FRN along the soil profile. It assumes an exponential depth distribution, and the depth of soil removed by erosion is estimated by comparing the reduction in the FRN inventory with that related to the reference site (see Porto et al., 2003). MODERN aligns the sampling site's total inventory to the measured shape of the reference site's depth profile to estimate the thickness of soil losses/gains. The intersection along the soil profile represents the solution of the model. We accounted for the uncertainty in the conversion procedure by running both conversion models 100 times, sampling from the reference and the erosion inventory within the uncertainty bounds and for the PDM in addition to the shape factor



$h_0$. The sampling was done from normal distributions, defined by the mean measured value and the
standard deviations (SD): i) of the repeated ICP-MS measurements for the erosional sites, ii) of the
replicate reference inventories, iii) of the three depth profiles for the $h_0$ factor (Supplementary Figure

139    1).

**2.4    Sediment yield measurements**
Since 1978, precipitation, runoff and sediment yield have been measured in the W2 catchment (Cantore
et al., 1994). Precipitation was recorded using a tipping bucket rain gauge, and runoff was measured at
the outlet using an H-flume structure equipped with a mechanical stage recorder. Below the H-flume,
the sediment load was measured with a Coshocton wheel sampler (Porto et al., 2003). Sediment yield
data used in this analysis is related to the period from 1978 to 1994 (Cantore et al., 1994) and from 2006
to 2013 (Fig. 2). However, due to the malfunctioning of the sediment sampling equipment in the
catchment during some events, direct measurements of total annual sediment yield values are not
available for all years. To account for these missing years, the corresponding sediment output was
estimated using the Arnoldus Index, for which long-term observations are available from the station of
Crotone located ca. 10 km distant from the study catchment (see Capra et al., 2017). These estimates
were then incorporated into the annual record of sediment yield (Fig. 2), and the sediment yield data
was extrapolated to cover the period 1963–2013, corresponding to the period captured by Pu-239+240
derived soil redistribution assessments.



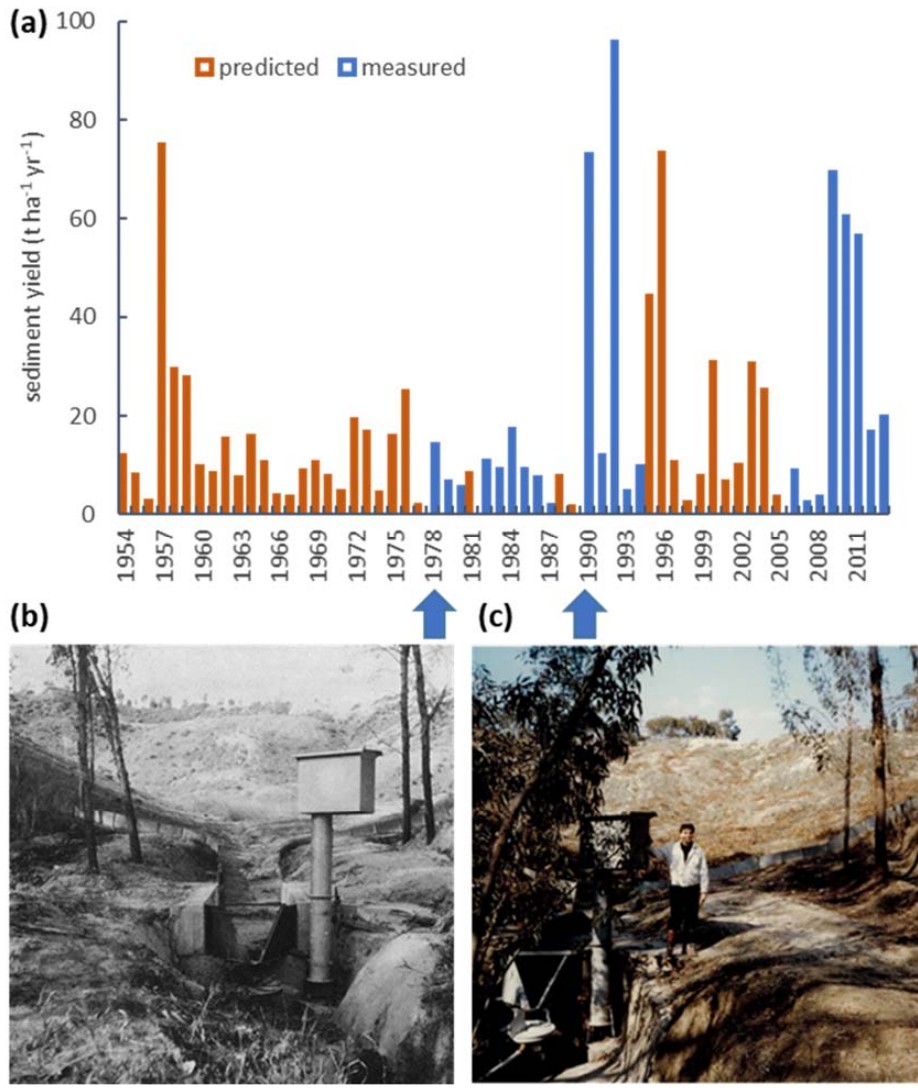


Fig. 2 Measured (orange) and predicted (blue) annual sediment yield (t ha-1 yr-1) of the headwater
catchment W2 (a). Predictions of sediment yields are based on a significant relation to Arnoldus
erosivity index. In 1968, Eucalyptus occidentalis Engl. was planted in the catchment that was harvested
in 1978 (b) (photo by M. Raglione, from Avolio et al., (1980)) and a second time in 1990 (c).

**2.5    Spatial extrapolation of Pu-239+240 derived estimates**
Regionalization of the point erosion estimates was done to compare the sediment yields measured at
the catchment's outlet with Pu-derived sediment yields. Therefore, generalized additive models (GAM)
were fitted to the measured erosion estimates. GAMs can account for nonlinear relationships by
coefficients that can be expanded as smooth functions of covariates. The smooth terms were modelled



by splines, and geographic coordinates (x and y) were modelled as a 2d spline. To prevent overfitting,
we used the restricted maximum likelihood (REML) method with the R package mgcv (Wood, 2006).
As spatial covariates, elevation, slope, aspect, flow accumulation, and scarce, discontinuous tree cover
(as 0 and 1 categorical variables, see Fig. 1) were tested. These covariates were derived from a DEM
with 3m spatial resolution using the terrain function from the raster package. Because of the small
sample size of 55 sites, only a maximum of three covariables could be added to the model. For cross-
validation (n=50) of the spatial prediction, the data were randomly split into 80% training and 20%
testing data.

## 172   3    Results and Discussion

### 173   3.1    Pu-239+240 distribution at the reference sites

The mean measured $^{240}$Pu to $^{239}$Pu atom ratios at the reference and sampling sites were 0.179±0.03.
These atom ratios corresponded to the atom ratio found for the global fallout (Kelley et al., 1999) and
confirmed global fallout as the sole source of Pu in the catchment.
The three reference depth profiles Pu-239+240 activity at the reference site showed different shapes
with soil depth (Supplementary Figure 1). While profile 2 displays the expected exponential decline
with soil depth, profile 1 shows signs typically expected from erosional processes and profile 3 of
depositional processes (Supplementary Figure 1). Therefore, only profile 2 was assessed to be suitable
for extracting the shape of the depth distribution for the conversion procedure (Fig. 3a). The penetration
depth of Pu-239+240 reached 205 kg m$^{-2}$, corresponding to 26 cm soil depth (Fig. 3b). With an
exponential model fit of the PDM, we derived an $h_0$ at 94 kg m$^{-2}$, representing the point where half of
the activity is stored. The mean Pu-239+240 reference inventory was estimated at 42.3+-3.5 Bq m$^{-2}$.
The fitted surface soil (0 cm) concentration was 0.45 Bq kg$^{-1}$ (Fig. 3a).



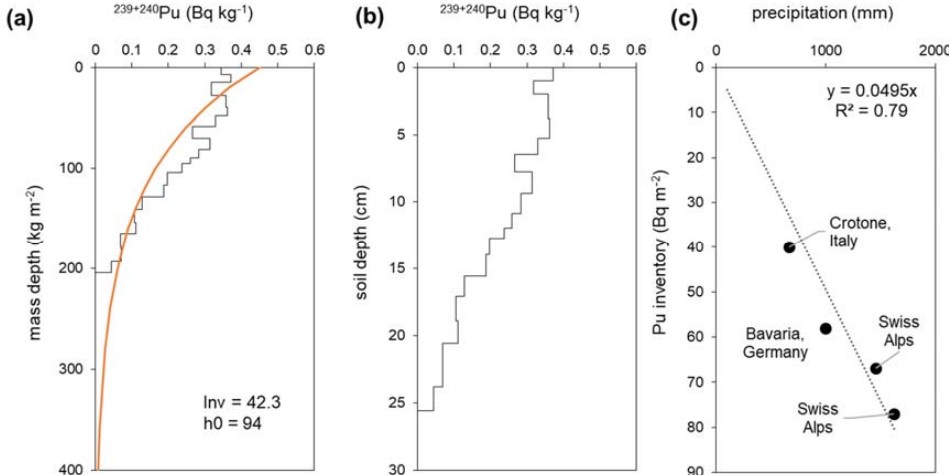


Fig. 3 (a) Pu-239+240 activity with soil mass depth measured at the reference site (selected profile 2). Inv corresponds to the total inventory of the soil, and $h_0$ to the shape factor of the exponential fit (orange). (b) Pu-239+240 activity with soil depth (cm). (c) Relation between Pu reference inventories and precipitation for studies in the Alps, Southern Germany (Schimmack et al., 2001; Alewell et al., 2014; Meusburger et al., 2018).

The Pu-239+240 measurements of the 19 bulk reference soil cores showed a bimodal distribution with
six high inventories clustering at a mean of 40.2 +- 4.4 Bq m$^{-2}$ and 13 low inventories of 15.0 +- 2.8 Bq
m$^{-2}$. The Pu-239+240 activities of the bulk soil cores with low inventories had values <0.043 Bq kg$^{-1}$,
close to the detection limit, and the standard deviation of replicate measurement of these samples was
high. To verify the plausibility of these low inventories, we calculated the Pu to Cs activity ratios. For
European soil samples, the activity ratio of Pu to Cs (with Cs being decay corrected to 2014) is expected
between 81 and 24 (Meusburger et al., 2020). However, the low inventory bulk cores had mean Pu/Cs
ratios of 156, which is clearly outside this range. A possible explanation for these very low Pu values
in the reference site might be due to the mixing and dilution of deeper layers with no Pu activity into
the bulk reference soil cores. Therefore, these low reference bulk samples were removed from further
analysis. Bulking of Pu samples causing a dilution of the Pu activity should be avoided, particularly in
areas of high erosion or low initial fallout (Wilken et al., 2021). Here, we were able to resolve the
dilution problem due to the availability of Cs-137 data, as the Cs-137 to Pu-239+240 activity ratios
were valuable in identifying the suitability of the reference samples. The plausibility of the Pu inventory
was further underpinned when the inventory was related to the mean annual precipitation of other
published European studies (Fig. 3c). The few published Pu inventories in Europe (Schimmack et al.,
2001; Alewell et al., 2014; Meusburger et al., 2018) show a linear relation to mean annual precipitation
with 77, 67, 58 Bq m$^{-2}$ for 1650, 1450, 950 mm of rainfall. The high inventory of this study of 40.2 Bq
m$^{-2}$ plots on the linear relation (Fig. 3c), while the low inventory of 15 Bq m$^{-2}$ is below the expected



amount given the catchment's mean annual precipitation. Taking the depth distribution reference and
only the six high inventories of bulk soil cores into account, the mean reference inventory of the soil
profiles was 40.5 +- 3.5 Bq m$^{-2}$ with a coefficient of variation of 8.6%.
All in all, following the above-described procedure, the Pu-239+240 reference inventories had a small
spatial variability with a CV of <9%. For Cs-137, the CV was 11.6% in the same reference area (see
Porto and Callegari, 2022). The spatial variability of Pu in reference sites was comparable to previous
studies (Alewell et al., 2014; Meusburger et al., 2016).
**3.2    Catchment inventories and soil redistribution rates at sampling points**
The Pu-239+240 activities at the sampling sites ranged from 0.001 to 0.143 Bq kg$^{-1}$ with a mean of
0.066 Bq kg$^{-1}$. The uncertainties of repeated ICP-MS measurements increase with decreasing activities
from the smallest SD of 0.0004 Bq kg$^{-1}$ to the largest of 0.067 Bq kg$^{-1}$ corresponding to <1% to larger
>100% of the measured activity with a median of 20%.
The respective mean Pu-239+240 inventories for all 55 sites were 16.8 Bq m$^{-2}$ with a spatial SD of
±10.2 Bq m$^{-2}$, thus less than half of the reference inventory. Given the uncertainty bounds, all
inventories, except for four sites, were significantly smaller than the reference inventory, indicating soil
erosion (Fig. 4a). One site close to the catchment outlet had a very high Pu-239+240 inventory of 111
Bq m$^{-2}$ exceeding the reference inventory by almost three times (Fig. 4a). The Pu-239+240 inventories
are significantly (p<0.001) correlated to the Cs-137 inventories with 24.7 times more Bq m$^{-2}$ for Cs-
137 (Fig. 4b). The Cs/Pu activity ratios of the catchment sites were at the lower range of the plausible
fallout range (between 23.9 = global and 81.3 = Chernobyl) with a mean value of 24.7. The activity
ratios are significantly (p<0.005) decreasing with decreasing erosion rates even though Rsq of the
regression is with 0.15 very low (Fig. 4c).
This depletion in Cs-137 pointed towards a preferential loss of Cs-137 during soil loss. A possible
explanation might be that Cs-137 is transported with different soil particles as Pu, which are more
susceptible to soil erosion. It is known that Pu-239+240 exhibits a different sorption behaviour to soil
particles compared to, e.g. Cs-137. Pu is mainly associated with organic matter and sesquioxides in
addition to clay particles, whereas Cs-137 is predominantly bound to the fine mineral clay fraction
(Lujaniene et al., 2002; Qiao et al., 2012; Meusburger et al., 2016; Xu et al., 2017). As a consequence,
Pu-239+240 is more exchangeable and might more easily migrate downward in soils (Schimmack et
al., 2001; Meusburger et al., 2016). This different sorption behaviour may result in different depth
distributions, which have important implications for its use as a soil erosion tracer, e.g. regarding the
conversion of measured FRN inventory changes into soil redistribution rates. Further, it may also have
implications regarding interpreting Cs-137 to Pu-239+240 activity ratios that may be shifted outside the
expected ranges at sites affected by soil redistribution.




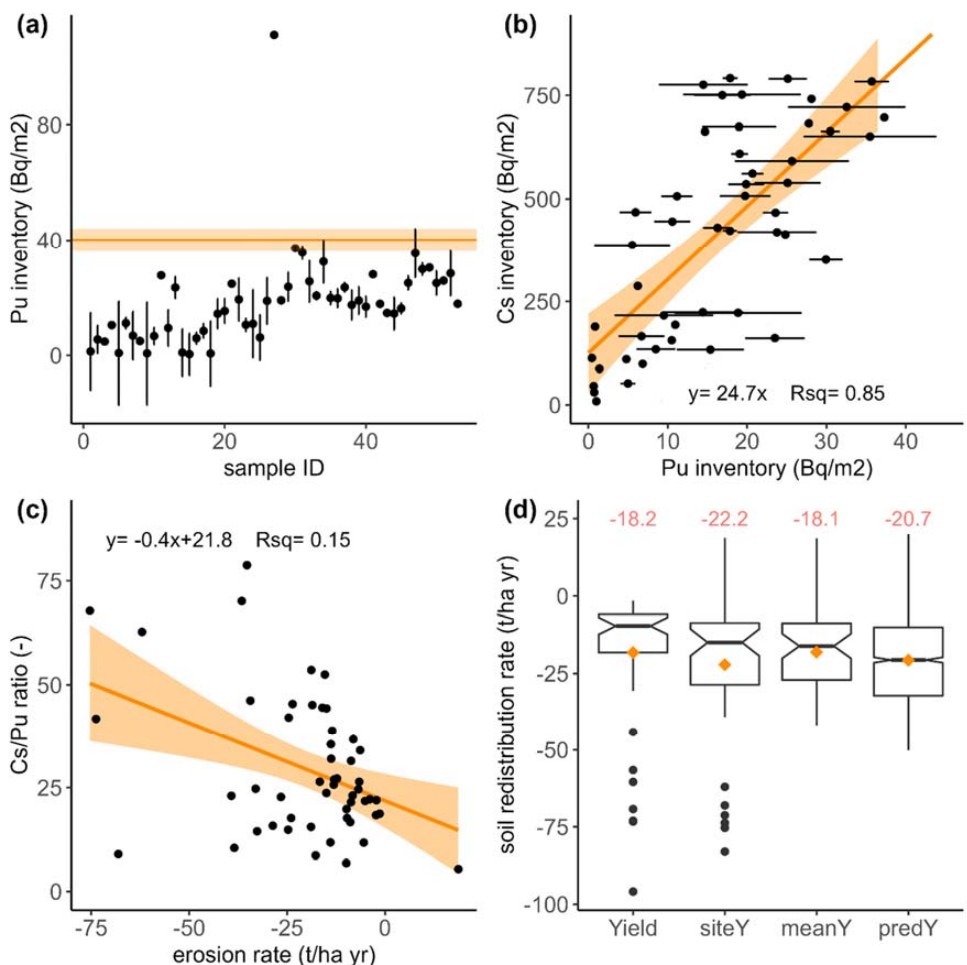


Fig. 4 (a) Pu-239+240 inventories with measurement errors in relation to sample ID (points) and the
reference inventory (orange line with ribbon). (b) Relation between Pu-239+240 and Cs-137 inventories
(error bars indicate the measurement error for Pu) with a linear trend line. (c) Activity ratio of Cs to Pu
versus erosion rate. (d) Measured sediment yield at catchment outlet (Yield), Pu-derived erosion rates
based on measured inventories within the catchment (siteY) and as a mean of the repeated conversion
results (meanY), and mean of regionalized catchment Pu-derived erosion rates (predY). Orange points
and text show the mean values of each approach.

**3.3    Comparison of Pu-239+240 derived soil redistribution rates and sediment yield of
       the catchment**

We produced three sets of Pu-derived soil redistribution rates using i) direct conversion of the site

inventories (siteY) and ii) the average of 100 Monte Carlo conversion models per site generated by

sampling within the uncertainty ranges of all input parameters (meanY) and iii) regionalized estimates





for the catchment (predY). For the point estimates will refer to these meanY in the following because
the uncertainty related to the entire procedure is included in this second set of redistribution rates.
Soil redistribution rates were highly variable within the catchment (Fig. 5). The highest soil loss with
$43 \pm 20$ t ha$^{-1}$ yr$^{-1}$ occurred in the upper part with patchy tree cover. Generally, the sites with scarce tree
cover and adjacent sites showed the highest soil erosion rates. Downslope and towards the outlet of the
catchment, the erosion rates decrease. Close to the outlet, soil deposition of $18.7 \pm 2.0$ t ha$^{-1}$ yr$^{-1}$ was
observed in one measurement point (W2_29). The deposition rate is, however, difficult to quantify
without knowledge of the respective soil source area or a Pu depth profile in the deposition site. The
average of all measured site redistribution rates (siteY) indicated erosion of -22.2 t ha$^{-1}$ yr$^{-1}$ with a spatial
standard deviation of $\pm 21.1$ t ha$^{-1}$ yr$^{-1}$. On average, the standard deviation, derived from repeated Monte
Carlo conversions, of these redistribution rates were 7.2 t ha$^{-1}$ yr$^{-1}$, with a slightly lower median of the
standard deviations of 4.2 t ha$^{-1}$ yr$^{-1}$ corresponding to a mean CV of 45% and a median CV of 36%.
Generally, higher erosion estimates are subject to higher standard deviations resulting from higher
uncertainties for measuring low Pu activities. Excluding these measurement uncertainties from the
Monte Carlo conversion reduced the CV of the erosion estimates to mean and median CVs of 19% and
13%, respectively.
The XY-coordinates, elevation, and flow accumulation best explained the spatial pattern of soil
redistribution rates. The deviance explained with these two spatial covariates was 56.7%, with lower
accuracy of 24% for the cross-validation procedure. The spatial pattern of the predicted soil
redistribution rates showed erosion in most of the catchment (Fig. 5). Only in grid cells with high flow
accumulation deposition occurred. The average redistribution rate from the grid cells (predY) amounted
to -20.7 t ha$^{-1}$ yr$^{-1}$ (Fig 4d). Given the measured sediment yield at the outlet (Yield) of -18.2 t ha$^{-1}$ yr$^{-1}$,
this corresponds to a 14% overestimation of soil loss by the Pu method (Fig 4d). The sediment yield
(Yield) corresponds to the off-site net erosion over time while the Pu-derived rates (siteY, meanY and
predY) to the on-site erosion over space. Their correspondence indicates that most of the on-site eroded
sediments are delivered to the outlet of the stream channel within the considered period.



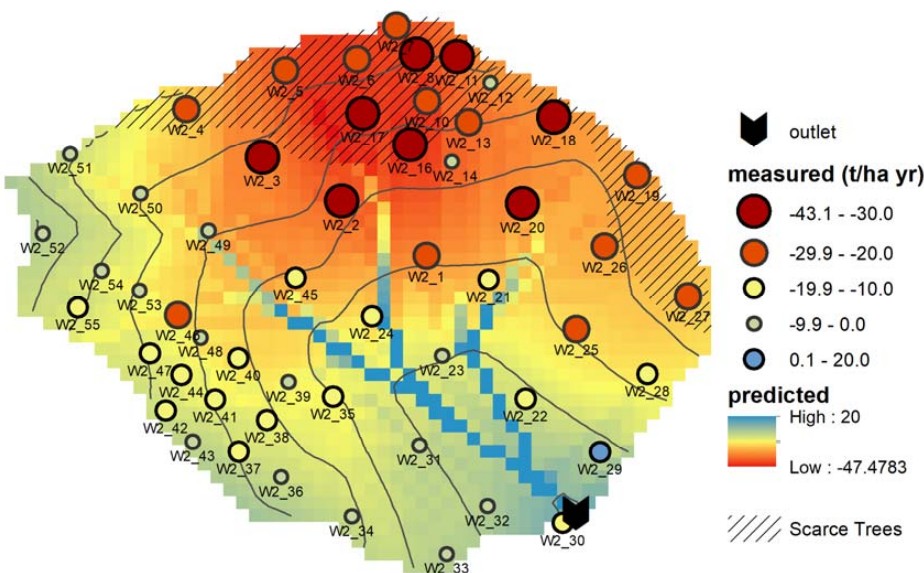

Fig. 5 Soil redistribution rates assessed with Pu-derived soil redistribution rates (points) and spatial prediction of soil redistribution rates based on these point rates using XY-coordinates, elevation and flow accumulation as spatial covariates.

The Pu-derived soil erosion rates in the catchment were very high, with maximum values <-40 t ha$^{-1}$ yr$^{-1}$. However, documented soil erosion peaks in this area can reach up to 100–150 t ha$^{-1}$ yr$^{-1}$ during exceptional rainfall events (Porto et al., 2018; Porto et al., 2022). The sediment yield time series reveals that besides the rainfall erosivity, particularly the second harvest of eucalyptus trees (1990), triggered soil erosion. The soil conservation effect of the eucalyptus trees was also revealed by the lower Pu inventory and, therefore, higher soil losses in the catchment area with scarce tree cover. The protective effect of trees (Sorriso-Valvo et al., 1995; Zhou et al., 2002) and vegetation cover, in general, was also found in other studies and reviewed by Zuazo and Pleguezuelo (2009). Flow accumulation, a proxy for runoff concentration in a catchment, was an important predictor of soil erosion patterns. Interestingly, the relationship was negative with lower soil losses and higher chances for deposition with increasing flow accumulation. A reason for this was likely the collinearity between decreasing slopes with increasing flow accumulation, reducing the sediment transport capacity (Xiao et al., 2017). Still, flow accumulation performed better than alternative GAM models, including slope.

Mean Pu-239+240-based mean soil redistribution rates were -20.7 t ha$^{-1}$ yr$^{-1}$ and 14% higher as measured sediment yields at the catchment outlet. Given both methods' uncertainties and variability, comparable magnitudes were achieved. In a recent study, Porto and Callegari (2022) found Cs-137 redistribution mean rates of -20.4 t ha$^{-1}$ yr$^{-1}$. The Cs-137 and Pu-239+240 derived soil redistribution rates are in good agreement.



## 4   Conclusion

Recent measurements of Pu-239+240 in a catchment in Southern Italy provided essential insights into the suitability of the Pu-239+240 technique to estimate soil erosion rates. We also rigorously tested the uncertainties involved in the approach. In our case study, the highest uncertainty resulted from the high measurement uncertainty of low inventory samples, with a median CV of 21% and high measurement uncertainty of <1% – 100%. This high uncertainty can, for future studies, be minimized by (i) taking incremental soil depth samples, avoiding dilution with deeper horizons of low-activity soil and (ii) extracting Pu on larger soil samples (~20g). Based on values with adequate measurement certainty, the Pu-239-240 technique showed a low spatial variability of the reference inventory (CV <9%) and the shape of the Pu distribution within the soil profile proved stable adsorption to the topsoil. Patterns of inventory loss were related to soil redistribution processes, with the best spatial predictors being tree cover and flow accumulation. The Pu-assessed redistribution rates were in agreement with Cs-137-derived rates and sediment yield measurements at the catchment outlet.

Increasing climatic extremes associated with more intense farming practices endanger our soil resources, and new tools to monitor soil losses are of utmost importance. So far, the tracer Cs-137 has been a powerful approach to assess soil redistribution rates since its fallout. However, alternative tracers are needed in light of the subsequent decay of Cs-137 approaching the detection limit. The Pu-239-240 technique works analogue to the Cs-137 technique. We conclude that Pu-239+240, with its considerably longer half-life, is a suitable and promising soil redistribution tracer.



## Data availability

The authors declare that all other data supporting the findings of this study are available within the article and its Supplementary Information files.

## Acknowledgements

We thank the University of Cadiz for the measurement of Pu-239+240 on the ICP-MS.

## Author information

### Contributions

K.M., C.A, and P. P. conceptualized the study. P.P. collected the samples, J.K.-W. measured them and calculated the measurement uncertainties. K.M. and P.P. did the data analysis. K.M. wrote the manuscript, and all co-authors contributed to the writing process.

### Competing interests

The authors declare no competing financial interests.





## Appendix

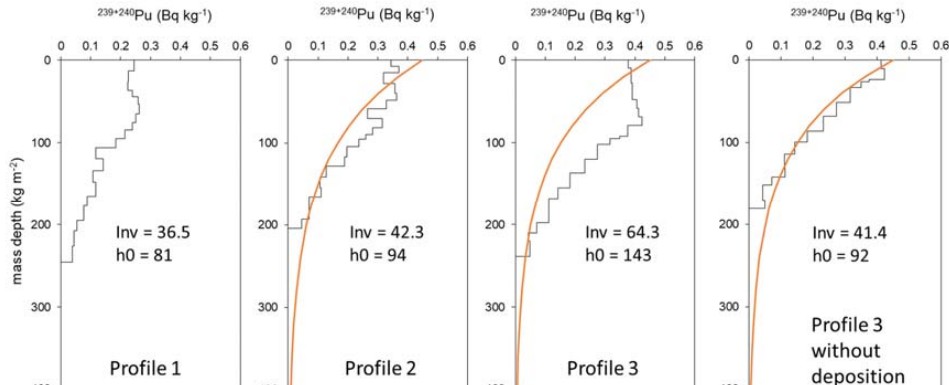

Figure A1: Pu-239+240 activity with soil mass depth measured at three potential reference sites. Inv corresponds to the total inventory of the soil, and $h_0$ to the shape factor of the exponential fit (orange). Profile 3 was fitted with and without deposition layers. The standard deviation of the depth distribution and $h_0$ factor of profiles 1, 2 and 3 (without deposition) was used for the uncertainty assessment in the conversion model.



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
