# Peer review of "Validating Plutonium-239+240 as novel soil redistribution tracer a"

_EGUsphere, 2022_

## Author Response (AR1)

**Review of manuscript EGUSPHERE 2022-1359**

*We thank both reviewers for the positive feedback on the manuscript and the additional points raised to improve it. Please find below a detailed reply on how we implemented the helpful changes suggested by the reviewers.*

**Reviewer 1**

**Recommendation:**

Publish with minor modifications.

**Major comments on methodological issues:**

1) In Chapter 2.1 there is no information on the slope inclination (and eventual microrelief) of the reference site (lines 97-100). This should be mentioned there.

*We added the information on the slope of the reference site in line 95:*

*"In 2014, the collection of soil samples in the catchment was undertaken along an approximate 20 m × 20 m grid with additional cores collected from areas characterized by marked variability of vegetation cover and topography **with slopes from 5 to 35° (Fig. 1)."***

2) Use of conversion models:

In chapter 2.3. It is mentioned that two conversion models were used (Profile Distribution Model of Walling and MODERN of Arata (lines 123-128) But further this does not have any implications on data processing. For example, in Chapter 2.5 when there is mentioned comparison of soil erosion rates obtained by Plutonium-239+240 with sediment yields at the catchment outlet (lines 160-171) it is not mention whether the results of PDM or MODERN were used for comparison. In fact, there is no use of results provided by two models. So, I suggest either: a) discuss which model gave higher or lower values, which model seem to be more reliable, for which model we can obtain the input parameters easier or more precise, and provide some recommendations on using these models, or if not, b) do not mention two models, select one which seems to be better and do not use and do not mention the other one in this paper. I think it would be a valuable information to compare the results of these two models, because MODERN is very new model and yet only very few people use it. Yet I am not aware of any study which would be using MODERN and Walling's models for the same data set and compare their performance (may be there exist such studies and I did not catch them up, but I presume even if they exist, they are very few. May be this is not right place to make such comparison in this paper because this does not match with the objective of this paper, but than it would be a good topic for another separate paper.

*Sorry for this confusion. We were also undecided between option a) and b). Thus, in a previous version we included the comparison of the model while in a later version we removed it due to make the manuscript more concise. As a compromise for the interested FRN community we moved the comparison now to the Supplementary Figure 2. Since we already provided a detailed comparison in the paper:*

*"Meusburger, K., Porto, P., Mabit, L., La Spada, C., Arata, L. and Alewell, C., 2018. Excess Lead-210 and Plutonium-239+240: Two suitable radiogenic soil erosion tracers for mountain grassland sites. Environ Res, 160: 195-202."*

3) There was mentioned that generalized additive models (GAM) were used (lines 161-163). I think this formulation is somehow vague and it is not clear what for GAM were used. May be this can be explained?

*We used the GAM for regionalization of the point soil redistribution estimates to the entire catchment this is needed to make the comparison with sediment yields. See line 174-175. Since this was not clear we rephrased the sentence and slightly restructured the respective paragraph.*

4) Ratios of Pu and Cs and PU-239 and Pu-240:

In Chapter 3.1 there is discussed ratio of different isotopes with respect to distinguishing bomb derived fallout and Chernobyl fallout (lines 174-176 and 196-201). If it is possible to use the rations of different isotopes to distinguish between bomb-derived fallout and Chernobyl fallout, it would be very useful. But please, explain this feature more in detail.

*This is a very good point we only mentioned the ratios very briefly in the introduction. Now we added a more detailed section on it in the methods (line 128-134):*

*"The information of $^{239+240}Pu$ in relation to $^{137}Cs$ allows for assessing the origin of the fallout (e.g., Chernobyl derived versus global bomb fallout). A prerequisite for using the $^{137}Cs$ to $^{239+240}Pu$ activity ratio is that Pu is exclusively derived from global fallout. Thus, in a first step, the $^{240}Pu$ to $^{239}Pu$ atom ratios around 0.18 confirmed the $^{239+240}Pu$ origin merely from global bomb fallout (Kelley et al., 1999). In a second step, $^{137}Cs$ to $^{239+240}Pu$ activity ratios reveal the percentage of bomb derived (ratio 0.027) versus Chernobyl derived (ratio 0.013) (Ketterer et al., 2004; Xu et al., 2013; Meusburger et al., 2016; Meusburger et al., 2020)."*

5) Plutonium inventories:

I am very surprised that the plutonium inventories are so low (lines 182-185 and 192-196). Is this usual also in other areas or it is just here?

*Within a European context the inventories are relatively low due to the low rainfall in our Mediterranean study site. The relation between rainfall and Pu inventories is shown in Figure 3 c for European studies. However, in global context even lower inventories might be observed. We added studies that provide this global context (line 232-234).*

6) Designation of Fig 3:

The designation of Figure b) should be: "Pu-239+240 activity with soil depth (cm) at reference site (profile 2)" to be clear that the figures a) and b) show the same profile but once calculated per soil mass and once per depth. The designation c) has wrong label, because there is written that Figure shows sites from Alps in Southern Germany but there are also sites from Italy and Switzerland.

*Thanks, we changed the label of b) according to your suggestion. For c) we replaced "sites from Alps in Southern Germany" with "European studies". This will also help to distinguish it from global studies (see previous reply).*

7) Heterogeneity of reference values:

The heterogeneity of reference values (Chapter 3.1., lines 192-196) can be caused also by the vegetation heterogeneity at reference site. In chapter 2.1 (lines 98-100) it is mentioned that there is a rangeland with scattered trees. This can result in huge FRN microvariability, because of heterogeneous fallout input (redistribution of rainfall by tree branches, shadowing some spots from rainfall and concentrating rainfall to other spots. This is shifting with time, because some trees can be cut or die during decades, young trees can grow, and completely new trees can germinate. Secondly, the scattered tree bush and grass vegetation created difference in soil temperature and moisture what results in local migration of burrowing animals and great soil bioturbation.

*Thanks. This is a point that merits attention when establishing the sampling campaign on a reference site. In our case, the sampling area shows a patchy tree cover with dominance of grassed areas. The trees (Quercus pubescens Willd) are natural, irregularly spaced, and they have never been cut since the commencement of fallout. We avoided sampling the areas covered by canopy to minimize the above effects. Each sampling point was carefully chosen in the clearing areas far from the tree trunks to avoid also problems due to stemflow. The heterogeneity of the reference values is something that should be expected in similar cases because other causes like burrowing animals and soil bioturbation (as the referee said) cannot be excluded for such long time-window (1954-2021). However, it must be recognized that, as we mentioned in the main text, the spatial variability (CV = 9%) is relatively low compared to published studies using FRN as soil erosion tracer and we feel confident that the mean reference value is representative of the area. We added more information on the selection of reference sites in line 104-107.*

8) Sources of uncertainty:

In Chapter 4 it is mentioned that major source of uncertainty is analytical error, because the concentrations are very low (lines 310-312). But in Chapter 2.4. 150-153 it is mentioned that the sediment yield measurements at catchment outlet had to be extrapolated to period 1963-2013 to match with the period represented by Plutonium method. This is obviously another source of uncertainty. How can we know how much uncertainty this can contribute?

*This is a very good point! Indeed, the temporal extrapolation of sediment yield is also a source of uncertainty. As described in the methods, a regression between Arnoldus index and sediment yield measurements was used for the extrapolation. The standard error of this regression was 23 t ha$^{-1}$ yr$^{-1}$. Also, the variation between sediment yield between years is very high, with 21.8 t ha$^{-1}$ yr$^{-1}$. Therefore, we added the information in the method description line 166 and discussion line 305. However, in Chapter 4, we solely conclude on the uncertainties of the Pu method. Therefore, we did not include it there.*

9) Reducing uncertainty caused by analytical error when measuring samples with low concentrations:

In chapter 4 it is written that the analytical error can be reduced by extracting Plutonium from larger samples (Line 314-316). Maybe it would be good to explain how much this can be

improved. I presume that larger sample would need more work, time, reagents so for how much we can improve it and how it would influence the time, labour, investment?

*Thanks for this comment! We investigated this in more detail by plotting the relative error against the Pu activities. We found relative errors of 7-22% for Pu activities <0.02 Bq kg$^{-1}$. The error declines to <2% for higher activities with a median of only 0.2%. Thus, if you have activities below this threshold, you can reduce your uncertainty by approximately one magnitude if your new extracts have activities above the threshold. However, if you have generally high activities, the effect of extraction of larger soil volumes will only marginally improve the already low relative error. We explained it in more detail in the text (lines 240-243) and added the graph to the Supplementary Figure 3.*

*Regarding the lab effort, slightly more effort and reagents are needed when extracting larger soil volumes.*

**Formal and language comments:**

Throughout the whole text there is used this form of isotope designation: Cs-137. However, this is more or less auxiliary alternative approach used mainly in older publications when printing facilities had difficulties with printing superscripts. Normally we use in scientific literature number in front of the element and in superscript, like this: $^{137}$Cs. Why authors used this alternative form? I think it would be better to use the form with number in superscript first and letters after.

*Thanks for pointing this out. We changed the writing throughout the manuscript. We further appreciate the detailed suggestions provided in an extra file.*

Reviewer 2

Really good work. With small corrections I suggest publishing. The research is innovative, expensive, and time-consuming. Here are some short comments:

Comment 1: Since the validation of the 239+340Pu method is the subject of this research, MDA (minimum detectable activity), measurement uncertainty of the method, repeatability, etc. should be evaluated.

*Thank you for pointing this out. The detection limit of a quadrupole ICP-MS (Q-ICP-MS) can vary depending on the specific instrument, operating conditions, and the element being analyzed. Typically, the MDA of a quadrupole ICP-MS is 10–100 fg per ml of extract (Alewell et al., 2017). We added this information to the manuscript. Given the specific activity of Pu-240, which is approximately 0.063 Bq/fg and 10g of soil used for extraction, this would correspond to 0.0063 to 0.063 Bq kg$^{-1}$. However, we found that the relative error increased significantly at a threshold of activities <0.02 Bq kg$^{-1}$. See the Figure below, which we added to the manuscript's Supplementary Figure 3.*

[Figure]

*Figure 1: Relation between Pu activity and relative error measures with a quadrupole ICP-MS.*

*Regarding measurement uncertainty and repeatability, we measured each sample multiple times to assess the SD of the measurement. These values have already been reported in the paper line 254-255. However, since the uncertainties are not normally distributed but are a function of the Pu activities, we added more information in line 241.*

Comment 2: How did you enter vegetation in the Pu/Cs inventories calculations?

*The vegetation cover is not considered in calculating the Pu/Cs inventories, as this should not influence FRN-based erosion rates. Instead, per definition, the FRN concentration and the mass depth of the soil are used to calculate inventories (line 136-138).*

*What needs to be considered, however, is that the reference soils should be under a similar vegetation cover as the erosional sites, as vegetation might shield deposition from soils. Further, the effect of the vegetation cover is implicitly considered when erosional inventories*

*are related to reference inventories. If vegetation cover has a protective effect against soil erosion, these inventories will be higher than inventories under scarce vegetation cover. Indeed, the vegetation cover was also a good covariate in the GAM model. However, since we had to restrict the number of covariates in the model, it dropped out.*

Chapter 2

2.1

line 93:  What kind of topographic variability is it? Maybe you could correlate topographic curvature values and slope values with plutonium/cesium inventories or erosion rates.

*We added the ranges of slope inclinations that we covered with our sampling, which ranged from 5 to 35° (line 95). Moreover, the elevation, aspect, and flow accumulation gradient in the catchment are considered in the spatial extrapolation.*

2.3

line 126: In the part where you mentioned PDM and MODERN I assume (according to the range of years of observed annual sediment yield) that you took 1963 as the year of the dominant FNR input? It should be explained in the text why exactly that year. Also, if there are any, list the programs you used for their conversions into erosion rates.

*Thanks for pointing this out! We added this information in lines 146-149: "We selected 1963 as the reference year for the erosion rate conversion. In 1963, the main global fallout peak occurred, commonly used in conversion models. The PDM equation was implemented in R, while MODERN was calculated with Matlab."*

2.3

line131: I assume that the erosive rates in the watershed are represented by the PD model. You wrote how MODERN was used for the thickness of soil losses/gains estimation. Where are these results?

*Thank you! Reviewer 1 raised the same point. Please refer to the detailed answer there.*

*In a previous version, we included the comparison of the model, while in a later version, we removed it to make the manuscript more concise. Now we moved the comparison to the Supplementary Information. Since we already provided a detailed comparison in the paper:*

*"Meusburger, K., Porto, P., Mabit, L., La Spada, C., Arata, L. and Alewell, C., 2018. Excess Lead-210 and Plutonium-239+240: Two suitable radiogenic soil erosion tracers for mountain grassland sites. Environ Res, 160: 195-202."*

Chapter 4

line 323 It must be emphasized the advantage of cesium in terms of sample preparation (it just dries and homogenizes), while with ICP-MS, you have to destroy the matrix and you perform chemical and other separations, which takes a long time and introduces errors.

This is a good point that we added (lines 347-350): "In most aspects, the concept of the $^{239+240}$Pu technique works analogue to the $^{137}$Cs technique. However, sample preparation with extraction is more demanding and destructive to the soil, while for the 137Cs method, soils are only sieved and dried and might be re-used for further analysis."